# Evaluating the Effect of a Neonatal Care Bundle for the Prevention of Intraventricular Hemorrhage in Preterm Infants

**DOI:** 10.3390/children8040257

**Published:** 2021-03-25

**Authors:** Maximilian Gross, Corinna Engel, Andreas Trotter

**Affiliations:** 1Department of Neonatology, University Children’s Hospital Tübingen, 72076 Tübingen, Germany; 2Center for Pediatric Clinical Studies, University Children’s Hospital Tübingen, 72076 Tübingen, Germany; Corinna.Engel@med.uni-tuebingen.de; 3Children’s Hospital and Center for Perinatal Medicine, Teaching Hospital of the University of Freiburg, 78224 Singen, Germany; Andreas.Trotter@glkn.de

**Keywords:** preterm infants, intracranial hemorrhage, prevention, neonatal care bundle

## Abstract

Germinal matrix intraventricular hemorrhage (IVH) remains a severe and common complication in preterm infants. A neonatal care bundle (NCB) was implemented as an in-house guideline at a tertiary neonatal intensive care unit to reduce the incidence of IVH in preterm infants. The NCB was applied either to preterm infants <1250 g birth weight or <30 weeks gestational age or both, and standardized patient positioning, nursing care, and medical procedures within the first week of life. A retrospective cohort study was performed to investigate the effect of the NCB and other known risk factors on the occurrence and severity of IVH. Data from 229 preterm infants were analyzed. The rate of IVH was 26.2% before and 27.1% after implementing the NCB. The NCB was associated neither with reducing the overall rate of IVH (odds ratio (OR) 1.02; 95% confidence interval (CI) 0.57–1.84; *p* = 0.94) nor with severe IVH (OR 1.0; 95% CI 0.67–1.55; *p* = 0.92). After adjustment for group differences and other influencing factors, amnion infection syndrome and early intubation were associated with an increased risk for IVH. An NCB focusing on patient positioning, nursing care, and medical interventions had no impact on IVH in preterm infants. Known risk factors for IVH were confirmed.

## 1. Introduction

Germinal matrix intraventricular hemorrhage (IVH) is a common form of brain damage in preterm infants <32 weeks of gestation [1]. IVH occurs mainly within the first days of life and can spread from the germinal matrix region into the lateral ventricles. If the bleeding obstructs terminal veins, periventricular hemorrhagic infarction (PVHI) can occur [2,3]. While even mild IVH (grade 1 and 2) may increase the risk of neurodevelopmental impairment, severe IVH (grade 3 and PVHI) is particularly associated with a poor neurological outcome [4].

Since there are few therapeutic options after diagnosing IVH [5], prevention is of crucial importance. Prenatal preventive measures include avoidance of preterm delivery, antenatal corticosteroid therapy, and transfer of mothers in preterm labor to centers specialized in high-risk delivery [6,7]. Postnatal preventive interventions aim to reduce stress and hemodynamic fluctuations through appropriate respiratory support and adequate blood pressure management. Increasing evidence suggests that patient posture, i.e., keeping the preterm infant’s head in midline and elevated position, decreases IVH incidence [2,8,9,10,11,12,13,14].

Based on average performance concerning (severe) IVH reported on a German nation-wide information portal on the quality of care for very-low-birth-weight infants [15], in 2012, the study site’s neonatal intensive care unit (NICU) introduced a neonatal care bundle (NCB) as part of an in-house guideline and quality improvement initiative aiming to reduce the incidence of IVH. It focused primarily on patient positioning and minimal handling during the first week of life and applied either to infants <1250 g birth weight or <30 weeks gestational age or both. The NCB encompassed several aspects for which associations with IVH have been reported or are thought to influence IVH risk: avoidance of head rotation and tilting the incubator to promote cerebral venous drainage [16,17,18]; specifying parts of nursing care and medical procedures (for example, using closed suction catheters and slow arterial blood sampling/flushing) in order to minimize rapid cerebral blood flow fluctuations [11,19]. The NCB was based on a treatment concept developed at the perinatal center of the Children’s University Hospital Ulm, leading to a reduction in IVH incidence [20].

To investigate whether the NCB affected the incidence of IVH, we conducted a retrospective cohort study.

## 2. Materials and Methods

### 2.1. Study Design

This retrospective analysis of two historical cohorts was carried out at the NICU of the Children’s Hospital Singen, a tertiary perinatal center. It covered a period between July 2006 and June 2017. The NCB was implemented in June 2012. Digital and paper patient records were analyzed. The study was registered at the German Register of Clinical Trials (trial no. DRKS00018859) and was approved by the ethics committee of the Albert-Ludwigs-University of Freiburg (application no. 350/19). Due to the anonymized data and retrospective design, the ethics committee granted a waiver of informed consent.

### 2.2. Patients

The analysis included all preterm infants with either <1250 g birth weight or <30 weeks gestational age or both admitted to the NICU within the first 72 h of life. Infants with complex congenital malformations (i.e., cardiac, renal, or neurological) or palliative care were excluded.

### 2.3. Routine Care and Neonatal Care Bundle for the Prevention of IVH

During the study period, respiratory care was performed according to international standards [21]. Preterm infants with respiratory distress syndrome received nasal continuous positive airway pressure or mechanical ventilation when necessary. The target range for CO_2_ was between 40 and 60 mmHg with a pH > 7.25. The oxygen saturation target was 89–95%. Surfactant was administered when FiO_2_ requirement was >0.3 to 0.4. Arterial hypotension, defined by mean arterial pressure lower than gestational age in mmHg combined with poor peripheral perfusion, was treated by administering isotonic fluid boluses or inotropic agents (dopamine, noradrenaline, or norepinephrine) as needed. Indomethacin and ibuprofen were used for pharmacologic closure of hemodynamically significant persistent ductus arteriosus (PDA). Fluid management followed the European Society for Paediatric Gastroenterology, Hepatology, and Nutrition guidelines on pediatric parenteral nutrition [22]. Prenatal management by the study site’s Obstetrics and Gynecology department consisted of tocolysis, antenatal corticosteroids, and treatment of amniotic infection syndrome (i.e., clinical chorioamnionitis) with antibiotics.

Before implementing the NCB, preterm infants were placed in varying positions, i.e., prone, supine, or lateral position, without explicit specifications for head positioning. Elevation of the bed’s head, i.e., tilting the incubator, was also not performed routinely. Nursing interventions and medical/diagnostic procedures were performed according to nurses’ and physicians’ discretion; drawing blood from arterial lines was not standardized.

After a three-month development and training period between March and May 2012, the NCB was introduced as an in-house guideline at the study site in June 2012. It standardized patient positioning, nursing care, and medical procedures within the first week of life (Table 1). The main focus was set on patient posture: (i) maintaining a supine midline position for the first three postnatal days, (ii) neutral head position with avoidance of prone position until the seventh day of life, and (iii) tilting the incubator to 10 to 20 degrees in the first week of life. Weight was checked on the first and fourth day of life; measurement of head circumference and length and routine nursing interventions such as cleaning the incubator and extended body wash had to be performed not earlier than the fourth day of life. All invasive medical procedures within the first week of life had to be performed by experienced neonatologists. Medical procedures and nursing measures had to be adapted to the care rounds; inadequate exposure to light and noise had to be avoided; signs of stress and pain were continuously monitored using eligible pain scales.

### 2.4. Neuroimaging

Before implementing the NCB, cranial ultrasound scanning (CUS) was performed on the first and fourth day of life. Afterwards, CUS had to be performed not earlier than the fourth day of life, except for diagnostic workup of severe anemia. In both cohorts, follow-up CUS was performed depending on respective findings at least once every two weeks until discharge. IVH was classified according to Volpe and Deeg [23,24] by three neonatologists.

### 2.5. Study Objectives

The study endpoint was the rate of IVH before and after implementing the NCB and investigating the influence of group differences in the cohorts and known/potential risk factors on IVH rate. Included influencing/risk factors were (i) treatment according to the NCB, (ii) amniotic infection syndrome, (iii) absence or incomplete antenatal corticosteroid therapy for fetal maturation, (iv) <28 weeks gestational age, (v) birth weight <1000 g, and (vi) arterial hypotension with the need for catecholamines.

### 2.6. Statistical Analysis

Basic patient data and the presence of risk factors for IVH were reported descriptively and stratified by the treatment according to the NCB (before vs. after). Possible differences between groups were evaluated using t-test in normally distributed numerical factors and Mann–Whitney U-test in non-normally distributed ones, as well as chi-square test for categorical items. Statistical significance was set at *p*-value <0.05. Statistical analyses were performed using logistic regression (outcome: occurrence of IVH) to adjust for group imbalances and influencing/risk factors for IVH factors (each coded yes vs. no depending on presence or absence) and generalized logit models (outcome: severity of IVH). Possible collinearity and interactions of the influencing/risk factors were tested. No adjustment for multiple testing was made. All *p*-values are descriptive. Analyses were performed using Statistical Package for Social Sciences Version 25 (IBM Corp., Armonk, NY, USA) and SAS Version 9.2 (SAS Institute, Cary, NC, USA).

## 3. Results

In total, 247 infants with either a birth weight <1250 g or <30 weeks of gestation or both were eligible for participation in this study. Eighteen cases met exclusion criteria (palliative care *n* = 9; chromosomal aberration with severe heart defects *n* = 5; patient charts not evaluable *n* = 2; antenatally diagnosed intracranial bleeding *n* = 2; admittance to the study site’s NICU >72 h of age *n* = 1), which led to 229 infants for further analysis. Two infants died before cranial ultrasound was performed. Respective data were reported but not considered as a competing endpoint in the analysis. Patient characteristics are shown in Table 2. Prophylactic indomethacin treatment for PDA and amnion infection syndrome was reported more often before implementing the NCB, while intubation within the first 72 h of life was reported more often afterwards. 

Implementing the NCB did not reduce the rate of IVH in general or its respective grades of severity (Figure 1). Preterm infants <500 g birth weight and <26 weeks gestational age showed higher numbers of IVH after the implementation, while a decrease in IVH rate was observed in the remaining groups stratified by birth weight and gestational age, except for groups 1000–1249 g and 28–29 weeks, respectively (Table 3a,b). In univariate analysis, treatment according to the NCB was not associated with a reduction in overall IVH (odds ratio (OR) 1.02; 95% confidence interval (CI) 0.57–1.84; *p* = 0.94). This was also the case for mild IVH (IVH grade 1 or 2: OR 1.0; 95% CI 0.7–1.44; *p* = 0.98) and severe IVH (IVH grade 3 or PVHI: OR 1.0; 95% CI 0.67–1.55; *p* = 0.92). Neither mortality (OR 0.95; 95% CI 0.39–2.28; *p* = 0.90) nor IVH-free survival (OR 1.06; 95% CI 0.60–1.86; *p* = 0.84) were affected by implementing the NCB. Using multivariable logistic regression to adjust for the reported group differences and risk factors for IVH, we included the influencing/risk factors mentioned in Section 2.5, and early intubation and prophylactic PDA treatment with indomethacin in the regression model. After adjusting for these factors, treatment according to the NCB was not associated with reducing overall IVH (adjusted (a) OR 1.0; 95% CI 0.42–2.2; *p* = 0.90). Amniotic infection syndrome and early intubation within the first 72 h of life were associated with an increased risk of suffering from an IVH (a OR 3.2; 95% CI 1.4–7.1; *p* < 0.01 and a OR 12.6; 95% CI 4.3–36.9; *p* < 0.01, respectively). The remaining influencing/risk factors showed *p*-values > 0.05 in the regression model.

## 4. Discussion

This retrospective analysis evaluated the effect of an NCB to reduce the incidence of IVH in very preterm infants. Implemented as an in-house guideline at the study site, the NCB aimed to reduce rapid cerebral blood flow fluctuations within the first days of life through standardizing patient positioning, nursing care, and medical interventions. We failed to detect any impact on both the overall rate of IVH and the respective grades of severity after implementing the NCB, as they remained virtually unchanged. We identified amniotic infection syndrome and endotracheal intubation within the first 72 h of life as risk factors for IVH, but the different incidences of these factors in our study cohorts before and after implementing the NCB did not explain the lack of effect of our NCB.

While the mean gestational age was the same in our cohorts, the mean birth weight was slightly lower after implementing the NCB (907.8 g vs. 952.3 g). After stratification by birth weight and gestational age, preterm infants <500 g and <26 weeks of gestation showed higher rates of IVH after implementing the NCB. Amnion infection syndrome and prophylactic indomethacin treatment for PDA were reported more often before implementing the NCB, while intubation during the first 72 h of life had to be performed more often afterwards. These discrepancies regarding potential influencing factors on IVH applied to the entire study population (Table 2) but were also true for the mentioned subgroup of preterm infants <500 g and <26 weeks (not shown). Whereas amniotic infection syndrome represents a detrimental factor and prophylactic indomethacin a potential protective factor in the pathogenesis of IVH [12,25], the need for early intubation may indicate more clinically unstable infants after implementing the NCB. Additionally, early intubation and mechanical ventilation are risk factors for IVH [26]. With the simultaneous presence of counteracting influential factors on IVH, we have no explanation for the higher IVH rate in the subgroup of preterm infants <500 g and <26 weeks after implementing the NCB other than this being a finding by chance related to the small sample size.

Apgar scores at five and ten minutes were lower after implementing the NCB. However, Apgar scores above six at five minutes were not associated with IVH in a recent cohort study [1]. Dalili et al. did not find significant associations between a low conventional Apgar score and IVH [27]. Furthermore, in Schmid et al., the cohort with reduced IVH rates had significantly lower five- and ten-minute Apgar scores [20]. As part of our statistical analysis, we included the five- and ten-minute Apgar scores in the regression model without altering the results (treatment according to the NCB: a OR for IVH 1.0; 95% CI 0.45–2.3; *p* = 0.94). Considering the already extensive model, we excluded Apgar scores from the final model. 

After adjusting for the mentioned group differences and influencing/risk factors using multivariable logistic regression, treatment according to the NCB was not associated with a lower (or higher) risk for IVH. We identified amniotic infection syndrome and early intubation within 72 h of life as risk factors for IVH at our study site, which is in concordance with other publications concerning this topic [26,28].

Besides the mentioned group differences, poor protocol adherence could have influenced the outcome as we did not quantify or record it. However, adherence to the NCB was assessed regularly during repeated daily ward rounds.

In 2013, Schmid and co-workers reduced IVH rates in preterm infants by implementing a comprehensive treatment concept and prospectively monitoring risk factors for IVH [20]. Our observed overall IVH rates of 26.2% (before NCB) and 27.1% (after NCB) are higher than those published by Schmid et al. regarding their cohort before introducing the treatment concept to prevent IVH (22,1%). However, in the cohort of Schmid et al., 23.2% of infants had a birth weight of 1250–1499 g, whereas this was only 8.2% in our cohort. The incidence of overall IVH and severe IVH for infants <1250 g was equal in our cohort (25.9% and 12.5%, respectively) and the cohort of Schmid et al. (25.3% and 11.9%). While the Schmid study included a much larger number of patients, the authors gave no explanation as to why the cohort after implementing the NCB was smaller (*n* = 191), and the recruitment period extended over a shorter time (23 months) than the cohort before NCB implementation (*n* = 265; 31 months). Although mean gestational age was significantly higher after implementing the NCB, Schmid et al. addressed this imbalance by adjusting for gestational age, which resulted in non-significant differences between the two cohorts concerning survival without IVH and severe IVH, especially in preterm infants with a birth weight <1000 g. The study authors also noted that interventions such as delayed cord clamping or cord milking might have influenced neonatal outcomes in the cohort after implementing the NCB (neither intervention was performed during our study). Likely, the effect of a particular treatment concept cannot easily be transferred to another center even though its elements have been adopted unchanged. In their publication, the authors pointed out that the center-specific measures can only be applied to other hospitals to a limited extent. Schmid et al. instead provided information on how a center with a high incidence of IVH could proceed to reduce the incidence of IVH, including interdisciplinary cooperation, identification of specific risk factors for IVH, comparison of the center’s approach preventing IVH with a center with a low IVH incidence, and development of an individual treatment concept and verification of its adherence [20].

Along with the work of Schmid et al., at least one other study has shown that similar interventions can lead to a decrease in IVH incidence. In 2019, de Bijl-Marcus et al. evaluated the effect of a nursing intervention bundle on IVH incidence, which was remarkably similar in its essential points to our NCB. The authors focused on posture (maintaining midline head position; tilting the incubator 15 to 30 degrees; avoiding head down positions and sudden elevation of the legs; avoiding rapid blood sampling from arterial lines; avoiding rapid intravenous/arterial flushes) during the first 72 postnatal hours. They observed a significant decrease in “new or progressive IVH” within the first 72 h after birth (a OR 0.34; 95%CI 0.20–0.56; *p* < 0.001) as well as in their primary composite outcome, “new or progressive IVH, mortality or cystic periventricular leukomalacia” (a OR 0.42; 95% CI 0.2–0.65; *p* < 0.001) [13]. These results are contrary to ours as De Bijl-Marcus and colleagues observed a decrease in IVH rate, especially in the group of very immature preterm infants, while we observed an increase. The cause of this discrepancy remains unclear since the two NCBs were quite similar in terms of their main components, but it may also be due to our smaller sample size. 

With one key point in patient positioning (i.e., head in elevated midline position) during the first days of life, the discussed NCB implemented pathophysiological considerations on the genesis of IVH into the clinical routine. Based on observations, the mentioned head position prevents venous congestion as a potential contributory cause of IVH [16,17,18,29]. Although this has been recommended for years [12], the clinical effect of specific patient and head positioning to prevent IVH in preterm infants during the first days of life is still under evaluation, with varying results [30,31,32,33]. Recently, Kochan et al. contributed to a possible clarification of this issue with a randomized controlled trial in which they found fewer PVHI in preterm infants when they were nursed in a supine position with the bed elevated at 30 degrees (which is considerably more than in our NCB) [34]. However, this study had several limitations as baseline characteristics differed among the groups, with more pre-eclampsia in the treatment group and more prolonged rupture of membranes in the control group. There was no difference in the overall rate of IVH and IVH grades of severity 1 to 3 [35].

Since the issue of optimal postnatal positioning is not yet conclusively clarified, it must also be taken into account that, for example, by strictly avoiding prone position during the first days of life, potential positive effects of prone position on ventilation and food tolerance are withheld from preterm infants [36,37,38].

This single-center retrospective cohort study has several limitations. The patient number may have been too small to detect an effect of the NCB on IVH, although no trend could be observed. Possibly, other factors crucial to the preterm infants’ care were not sufficiently analyzed by this study. There may have been lingering changes in care practices that we were not aware of but may explain the lack of effect of the NCB on the IVH rate. Concerning the logistic regression interpretation, it has to be considered that, inherently, several influencing/risk factors were concomitantly present in our cohorts, despite our testing for collinearity and interactions. For example, prophylactic indomethacin for PDA was typically applied only to very immature infants, mainly intubated within 72 h of life, while inherently having the highest risk for IVH. These factors may have led to non-significant results for obvious risk factors for IVH, such as low gestational age, when applying the full regression model. To reduce stress, and as CUS during the first three days of life has no impact on treatment (exception defined in Section 2.4), we performed the first CUS on the fourth day of life. Therefore, this study could not evaluate the NCB’s effect on the occurrence of IVH during the first three days of life. Our approach also hampered differentiation between IVH and prenatally occurring intracranial hemorrhage. However, the latter is relatively rare and should have had little impact on the detected IVH rates [39]. Transferring NCBs that have successfully reduced the rate of IVH elsewhere does not guarantee the same results in another center. Therefore, as with every intervention, the proof of concept always requires additional studies. As our study is, like others, not randomized, confounding factors for this multifactorial-triggered morbidity, so far unrecognized, may explain the lack of an effect of the NCB in our center. We decided not to further adhere strictly to the NCB but to apply more individualized care and treatment.

## 5. Conclusions

Implementation of an NCB focusing on patient positioning and minimal handling was not associated with a reduction in IVH rate in a mid-sized German NICU. Known risk factors for IVH were confirmed. Regarding the effectiveness of NCBs to reduce IVH, our study’s result is in contrast to the results of others [13,20]. It is conceivable that transferring a successful strategy from one NICU to another does not necessarily lead to similar results as local conditions may require a more individualized approach. Thus far, there is no convincing evidence from randomized controlled trials that special care bundles affect IVH outcome. All studies, including ours, have only a hypothesis-generating character and have no confirmatory power. Therefore, as IVH is one of the major morbidities in very preterm infants, larger, multicenter, randomized, controlled trials are needed to evaluate the discussed measures’ effectiveness to reduce IVH.

## Figures and Tables

**Figure 1 children-08-00257-f001:**
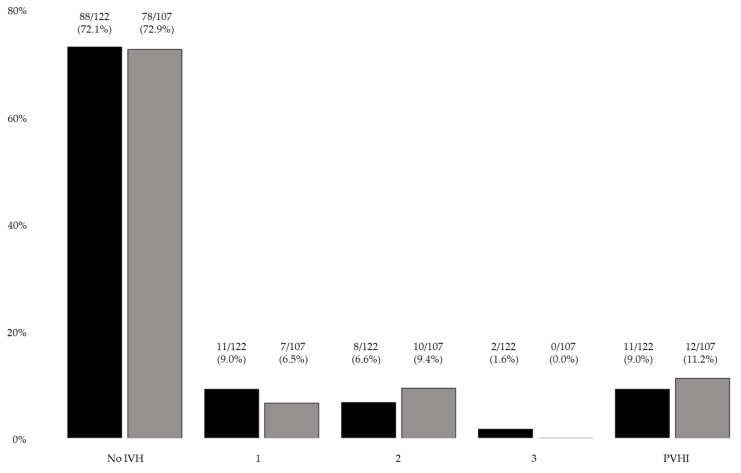
Number and percentage of IVH grades of severity; before (black) vs. after (grey) implementation of the neonatal care bundle (IVH—germinal matrix-intraventricular hemorrhage; PVHI—periventricular hemorrhagic infarction).

**Table 1 children-08-00257-t001:** Neonatal care bundle for the prevention of IVH (based on [20]).

**Patient Positioning**
- Tilt the incubator to achieve 10 to 20 degrees upper body elevation during the first week of life. - Supine midline position for three days with neutral head positioning. - Avoidance of prone position during the first week of life.
**Nursing Care**
- Nursing care and medical procedures should be combined and adapted to the infant’s sleep–wake cycle. - Apply individualized nursing care. - Always question the necessity of care procedures. - With six care rounds per day, time intervals can be chosen between three and five hours. - Nursing care during the first week of life has to be performed by experienced nurses.
**Care Procedures**
- Administer glucose–glycerol enemas every 12 h during the first three days of life. - Use closed suction systems on mechanically ventilated infants. - Measure head circumference and body length on the fourth day of life. - Measure weight on admission, then on the fourth and seventh day of life. - Avoid extensive cleaning of the incubator and body wash during the first week of life. - Perform the change of linen and measurement of weight with two nurses.
**Stimulation/Pain Management**
- Use of individual tactile stimulation during nursing care. - Avoid stress and pain. Evaluate stress and pain using pain scales.
**Light and Sound Environment**
- Avoid constant light exposure by covering the incubator. Visual monitoring must be provided. - Avoid noise by setting alarm tones as quietly as possible. - Acknowledge alarms quickly. - Do not place objects on the incubator. - Avoid noisy conversations near the incubator.
**Medical Procedures**
- Endotracheal intubation has to be performed by neonatologists/senior physicians during the first week of life. - Drawing of blood samples from arterial lines with subsequent flushing should be performed slowly (1.5 mL/30 sec) to avoid blood pressure fluctuations.
**Cranial Ultrasound**
- The first CUS has to be performed on the fourth day of life. - CUS has to be performed only by experienced physicians during first week of life. No routine doppler sonography.
**Review of Practice**
- All cases with IVH ≥ grade 3 and PVHI should be discussed within case conferences.

IVH—germinal matrix intraventricular hemorrhage; CUS—cranial ultrasound; PVHI—periventricular hemorrhagic infarction.

**Table 2 children-08-00257-t002:** Basic patient data and risk factors for IVH.

	Before Implementing the NCB to Prevent IVH (*n* = 122)	After Implementing the NCB to Prevent IVH (*n* = 107)	*p*-Value
Gestational age; mean (weeks)	27.9 ± 2.5	27.8 ± 2.8	0.67
Gestational age; median (weeks)	28.1 (21.9–32.9)	28.0 (22.0–36.1)	
Birth weight; mean (g)	952.3 ± 268.5	907.8 ± 285.6	0.23
Birth weight; median (g)	968 (320–1490)	920 (360–1480)	
Male	56/122 (45.9%)	53/107 (49.5%)	0.58
Small for gestational age	32/122 (26.2%)	31/107 (29.0%)	0.64
Multiple births	42/122 (34.4%)	39/107 (36.5%)	0.75
Caesarean section	112/122 (91.8%)	98/107 (91.6%)	0.95
Chest compression during resuscitation	1/122 (0.8%)	1/107 (0.9%)	0.93
Apgar 1; median	5 (1–9)	6 (1–10)	0.47
Apgar 5; median	8 (2–10)	7 (2–10)	<0.01
Apgar 10; median	9 (0–10)	8 (2–10)	0.04
pH at birth; median	7.32 (6.95–7.46)	7.32 (7.0–7.45)	0.59
Endotracheal intubation within 72 h of life	53/122 (43.4%)	61/107 (57.0%)	0.04
Overall intubation rate	62/122 (50.8%)	64/107 (59.8%)	0.17
Use of nCPAP during first 72 h of life	105/122 (86.1%)	92/107 (85.9%)	0.99
Treatment with Vancomycin/Cefotaxime	60 (49.2%)	41 (38.3%)	0.10
Treatment with Vancomycin/Meropenem	47 (38.5%)	40 (37.4%)	0.86
NEC ≥ 2 Bell stage	4/122 (3.3%)	1/107 (0.9%)	0.23
PVL	2/122 (1.6%)	1/107 (0.9%)	0.64
Prophylactic indomethacin for PDA	18/122 (14.8%)	2/107 (1.9%)	<0.01
Pharmacological PDA treatment	18/122 (14.8%)	21/107 (19.6%)	0.33
PDA ligation	5/122 (4.1%)	1/107 (0.9%)	0.14
ROP treatment	10/122 (8.2%)	6/107 (5.6%)	0.43
Discharge with supplemental oxygen	11/122 (9.0%)	13/107 (12.0%)	0.45
Length of NICU stay (days)	67 (1–164)	62 (1–160)	0.15
Mortality	12/122 (9.8%)	10/107 (9.4%)	0.90
**(Additional) risk factors for IVH**			
No antenatal corticosteroid therapy	15/122 (12.3%)	15/107 (14.0%)	0.47
Amnion infection syndrome	46/122 (37.7%)	22/107 (20.6%)	<0.01
Arterial hypotension	82/122 (67.2%)	66/107 (61.7%)	0.38
Use of inotropic agents	46/122 (37.7%)	39/107 (36.5%)	0.84
Pneumothorax	14/122 (11.5%)	11/107 (10.3%)	0.79
Neonatal transport	1/122 (0.8%)	2/107 (1.9%)	0.49

Data are presented as mean (± standard deviation); median (minimum and maximum) or as the number of patients with percentages in parenthesis. NCB — neonatal care bundle; IVH—germinal matrix intraventricular hemorrhage; nCPAP—nasal continuous positive airway pressure; NEC—necrotizing enterocolitis; PDA—patent ductus arteriosus; PVL—periventricular leukomalacia; ROP—retinopathy of prematurity; amnion infection syndrome—defined by presence of: (i) maternal fever >39 °C or >38 °C on repeated measurements and (ii) fetal tachycardia, (iii) maternal leukocytosis, (iv) purulent or malodorous discharge.

**Table 3 children-08-00257-t003:** IVH and survival rate classified from birth weight and gestational age before and after implementing the NCB.

	*n*		IVH Rate		Severe IVH or Death before CUS		Survival		Survival without IVH	
	*n*		Percentage		Percentage		Percentage		Percentage	
	Before	After	Before	After	Before	After	Before	After	Before	After
**(a)**										
<500 g	10	12	30.0%	58.3%	20.0%	41.7%	60.0%	50.0%	40.0%	33.3%
500–749 g	20	25	45.0%	36.0%	15.0%	12.0%	90.0%	92.0%	50.0%	56.0%
750–999 g	41	31	26.8%	19.4%	14.6%	9.7%	87.8%	93.5%	70.7%	80.6%
1000–1249 g	41	31	14.6%	22.6%	7.3%	3.2%	97.6%	100.0%	82.9%	77.4%
1250–1499 g	10	8	30.0%	0.0%	10.0%	0.0%	100.0%	100.0%	70.0%	100.0%
<1000 g	71	68	32.4%	32.4%	15.5%	16.2%	84.5%	85.3%	60.6%	63.2%
<1250 g	112	99	25.9%	29.3%	12.5%	12.1%	89.3%	89.9%	68.8%	67.7%
<1500 g	122	107	26.2%	27.1%	12.3%	11.2%	90.2%	90.7%	68.9%	70.1%
**(b)**										
<24 w	11	8	36.4%	62.5%	27.3%	37.5%	63.6%	50.0%	45.5%	25.0%
24–25 w	16	18	56.3%	61.1%	25.0%	27.8%	81.3%	77.8%	37.5%	27.8%
26–27 w	26	27	34.6%	22.2%	19.2%	11.1%	96.2%	96.3%	65.4%	77.8%
28–29 w	45	31	13.3%	22.6%	6.7%	3.2%	91.1%	96.8%	80.0%	77.4%
≥30 w	24	23	16.7%	0.0%	0.0%	0.0%	100.0%	100.0%	83.3%	100.0%
<28 w	53	53	41.5%	41.5%	22.6%	20.8%	84.9%	83.0%	52.8%	52.8%
<30 w	98	84	28.6%	34.5%	15.3%	14.3%	87.8%	88.1%	65.3%	61.9%
Total	122	107	26.2%	27.1%	12.3%	11.2%	90.2%	90.7%	68.9%	70.1%

NCB —neonatal care bundle; IVH—germinal matrix intraventricular hemorrhage; CUS—cranial ultrasound; w—weeks.

## Data Availability

The data presented in this study are available on request from the corresponding author.

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
