# Peer review of "Evaluating the Effect of a Neonatal Care Bundle for the Prevention of Intraventricular Hemorrhage in Preterm Infants"

_children, 2021, doi:10.3390/children8040257_

Round 1

Reviewer 1 Report

In this single center study, the authors describe the implementation of their neonatal care bundle to decrease the rates of IVH in infants < 1250 grams. The authors did not find any changes after implementation. The study report would be improved by having more unit specific descriptions of why they chose to implement the bundle, what are there hypotheses behind the results, and what lessons did they learn as a unit. It would also be a more robust discussion to have more comparisons with other studies.

Overall:

  1. Please fix “1.250 g” to either “1250 g” or “1.25 kg” – otherwise it is unconventional to most readers.

Introduction:

  1. What was the motivation to do this? Did you join a QI collaborative? Were rates already high at your institution?

Method:

  1. Can the authors describe why they chose 1250 grams as the cut off? Often it has been more customary to use <1000 grams (ELBW) or <1500 grams (VLBW)
  2. Define amniotic infection syndrome
  3. Lines 114-116 “Therefore, a p-value above or below 0.05 has no relevance but may inhibit the view on whether an observed difference of the groups may be of medical relevance” – I am not sure that this is accurate as it is customary to compare the two in order to provide readers with a quick reference to see whether there might be relevant significant differences between the two groups. In particular, the authors refer to in the discussion that the latter group had increased intubation rates and was thus sicker – therefore there is a descriptive comparison between the groups. In my opinion, it is helpful to provide the p-value and then readers can come to their own conclusions.

Results:

  1. Tables should have periods instead of commas for the percentage.
  2. Please provide p-values in table 2
  3. In Table 2 list of acronyms, Patent ductus arteriosus is spelled incorrectly.

Discussion:

  1. Higher rates of IVH after implementation of NCB? Hypotheses?
  2. What other hypotheses are there for the lack of change? What other practice changes were happening concurrently? There is a small reference to this in the discussion but perhaps the authors can expand on the ideas they have behind this.
  3. There is certainly data on expected rates of IVH. How does the unit compare?
  4. What is the significance of this study to other centers? What were lessons learned from this implementation? Has your unit continued to use the NCB in spite of these results?

Reviewer 2 Report

In this retrospective study the effect of a neonatal care bundle (NCB) on the incidence of Intraventricular hemorrhage (IVH) and severe IVH in very preterm infants was assessed.

Investigators calculated incidences in a cohort of very preterm infants, born before (2006-2012) and after (2012 – 2017) the bundle was implemented.

They did not find a difference in IVH incidence before or after the implementation of the bundle. In the tiniest infants (< 500 g) they even found an increased incidence, while in most other birthweight groups they found a slight decrease.

In general the paper is well written and reads well.

There are, however, some major flaws that need to be addressed:

  1. Patients: Were all the patients in-born?
  2. Methods: In the NCB group cranial ultrasound was performed not earlier than the fourth day, while in the pre-NCB group this was performed on the first and the fourth day. This may certainly have influenced the results. Small IVH’s may have been missed in the NCB group and the same is true for evolution of small to severe IVH’s. In addition, IVH occurring antenatally or during birth and thus not influenced by application of a NCB were not diagnosed in the NCB-group.
  3. Statistical analysis page 4: no p-values were calculated for basic patient data and IVH rates. This was not an RCT or matched control, but a retrospective study. Therefore, there may well have been differences in baseline characteristics between the two cohorts (before and after NCB implementation) that may have influence the results
  4. Statistical analysis page 4: The sentence “Risk-factors that showed a p-value <0.05 in the primary univariate analysis…” seems very unlogic, considering the first paragraph on statistical analysis  

I therefore feel the statistical methods should be checked by an epidemiologist and/or statistician and afterwards all calculations should be repeated, using the correct statistical analyses.

  1. Discussion: authors state that the higher need for intubation in the post-NCB group could not have influenced the results, as intubation was done by a skilled person. However, the need for intubation generally implies more severe respiratory illness which is related to IVH in preterm neonates. Therefore, this statement doesn’t seem to hold true.
  2. Discussion: De Bijl-Marcus et al found a significant improvement of ICH incidence after implementation of a NCB, also (and especially) in the most immature infants. Authors do not speculate on the important differences in the results between their study and the study done by de Bijl-Marcus et al. This is necessary for a good discussion.
  3. Discussion: cranial ultrasound was not performed in the first three days of life to reduce stress. I don’t agree with the authors: a cranial ultrasound examination, performed by a skilled person, aided by a neonatal nurse hardly causes any stress in the neonate. It would have given us more insight in the true effect of the NCB

Reviewer 3 Report

In this retrospective study, the authors analyzed the patient records on large cohorts before and after the implementation of NCB to investigate its efficacy on IVH in preterm infants. They could not find any significant impact on the severity of IVH. They also saw the known risk factors associated with IVH even after NCB.

Strengths

  • Large sample size (247), after exclusion 229
  • Long term study period, almost equal for before (6 years) and after (5 years)

Limitations

  • Discussion about breastfeeding, diet and antibiotic usage may be helpful
  • The study was done in only one center

Major comments

  • None that can question the article’s validity or importance

Minor comments

  • Little more on the effect of IVH and how the measures in NCB could help with its outcomes, in introduction may be helpful
  • Line 43, line 59, line 127= <1.250 g may be wrong, may be <1250 g or <1.250 kg
  • Illustrations of positioning would be useful

Reviewer 4 Report

In this manuscript, Gross et al. describe their experience with the implementation of a clinical care bundle to decrease the incidence of IVH in preterm infants at a single center. The authors do not find a difference with or without NCB.  Efforts to reduce IVH are important for the optimal care of the preterm infant and the authors should be commended for undertaking this important effort.  There are, however, significant problems with the statistical design of the study which ultimately make it unclear if the analysis is sound.

Is this a QI project?  The arrangement the authors describe what appears to be a QI project (implementation of care bundle after extensive education, comparison to historic cohort) without ever using those words.  On the other hand, it was registered as a clinical trial but with waiver of consent due to retrospective design.  This inconsistency needs to be rectified.

What was the compliance with the bundle?  From the introduction, many of these changes were quite different from current practice.  How often was the full bundle implemented (or was this a source of the lack of difference?)

The time period covered in the study is extremely long, nearly 11 years.  It seems implausible to this reviewer that other NICU practices remained consistent during such a long time period.  Although the mean gestational age was similar between the two groups, was there a difference in the number of babies born at 22 and 23 weeks between the groups (an exceptionally high IVH risk group)?

Table 1 is very difficult to read and needs to be reformatted.  The bundle items can be further subcategorized (nursing care, stimulation, positioning, medications, care procedures).

I am not convinced about the avoidance of describing differences in the control and intervention cohort.  I understand the concern about ascribed differences to random chance, but there are several very important differences.  In particular, prophylactic indomethacin, invasive ventilation, chorioamninonitis are quite different between the two groups and could have worked to opposite effect of the proposed intervention.

Was any power analysis performed to assess if the sample size was adequate to achieve the expect difference in IVH rate?

The multivariate modeling strategy is unclear to me.  What was the goal?  I would have expect to see a model where the effect of NCB on the binary outcome of IVH/no IVH (or severe IVH vs. no severe IVH) is evaluated controlling for important clinical variables as described above.  Instead the authors provide a model showing that treated hypotension and chorio are predictive of IVH.  While this is true, it is not really relevant to the project.

A better statistical strategy would be to match the control and NCB cohort with something like propensity scores to achieve parity across IVH risk factors, then to assess the impact of NCB.

Round 2

Reviewer 2 Report

The paper has improved significantly and most of my concerns have been addressed.

I still have some minor  and also more major issues.

Major: there were significant differences between the pre- and post NCB implementation groups. Not only concerning Indomethacin and need for ventilation, but also Apgar scores after 5 and 10 minutes. This may have influenced the results and still needs to be addressed.

Major:

Conclusion: I think a paragraph on the conflicting results between studies and on further study is warranted here. IVH remains a major complication of preterm birth and there seems to be a need for a larger, prospective, multi-center study and/or a large RCT. Benchmarking may also play a role in reducing the incidence of IVH

Minor:

  1. Abstract: NCB was applied too...(add: was)
  2. Introduction line 28: terminal veins instead of cerebral veins
  3. Introduction line 29: may increase instead of increase (as evidence on this subject is conflicting)
  4. Throughout the abstract and manuscript: <1250 g birth weight and/or < 30 weeks
  5. M&M line 71: respiratory care for all preterm infants... was conducted (or performed) according to....
  6. M&M line 83: omit triple I as this term is not commonly used amongst neonatologists
  7. M&M line 105-106: explain why CUS was only performed on postnatal day 4 in the post-NCB group
  8. Table 1: achieve instead of archive
  9. Table 2: see major comment 
  10. Discussion line 188: very preterm infants
  11. Discussion Line 204-214: it is not clear to me what is meant by the "mentioned subgroup", please explain

Reviewer 4 Report

I appreciate the opportunity to review the revision of this manuscript by Gross et al.  The reviewers have made a number of changes to the manuscript which have strengthened it, but several open questions remain inadequately addressed in my opinion.

I have two primary concerns.  First is about compliance with the bundle.  The entire focus of this study is the efficacy of the NCB to reduce rates of IVH, with the conclusion that it does not.  The reader is told that the compliance with the bundle was "mandatory" and "excellent" compliance was achieved, but this is never quantified.  As with any QI project, continual compliance with new clinical care practices is the primary challenge to overcome.  It is implausible that perfect compliance was immediately achieved and sustained over the course of the study period.  I would strongly suggest that the authors provide quantitative assessment of bundle compliance over the study period.

My second significant concern is the statistical approach.  I continued to be confused about the approach to the multivariate modeling.  The hypothesis of this study is that NCB reduces the risk of IVH.  Although I recognize that this study has a negative finding, I am perplexed why NCB was removed from their final model.  The goal of the modeling should not be to identify the best model for predicting IVH, but instead to look at the impact of NCB when controlling for important covariates.  By that measure, all models should have NCB forced into them, and covariates should not be chosen solely on their statistical significance but rather their known risk of contributing to IVH.  It is also problematic that important differences between the group (indomethacin and invasive ventilation) are not accounted for in the "final" models.

A continued minor concern is the assertion that no other changes were made to care practices between 2006 and 2017.  This also seems implausible as there have significant changes in the approach to non-invasive ventilation, feeding practices, and antibiotic regimens during this time period.  Could the authors please provide information on ventilator days, central line days, and length of stay to support the similarity between groups?
